**Data Availability Statement:** The dataset is publicly available using the following DOI: 10.6084/m9.figshare.12659813

# Survival among antiretroviral-experienced HIV-2 patients experiencing virologic failure with drug resistance mutations in Cote d'Ivoire West Africa

Boris K. Tchounga[1¤a]*, Charlotte Charpentier[2], Patrick A. Coffie[3], François Dabis[4¤b], Diane Descamps[2], Serge P. Eholie[3], Didier K. Ekouevi[4¤c]

1 Programme PACCI, Site de recherche ANRS de Côte d'Ivoire, Abidjan, Côte d'Ivoire, 2 Université de Paris, INSERM UMR 1137 IAME, Paris, France, 3 Département de Dermatologie et Maladies Infectieuses, Université Félix Houphouët-Boigny, UFR des Sciences Médicales, Abidjan, Côte d'Ivoire, 4 Centre Inserm 1219 & Institut de Santé Publique d'épidémiologie et de développement, Université de Bordeaux, Bordeaux, France

¤a Current address: Elizabeth Glazer Pediatric AIDS Foundation Yaoundé Office, Yaounde, Cameroon
¤b Current address: Agence Nationale de recherche sur le SIDA et les Hépatites, Paris, France
¤c Current address: Département de santé Publique, Université de Lomé, Faculté des Sciences de la santé, Lomé, Togo
* boris.tchounga@yahoo.fr

## Abstract

### Introduction

The long-term prognosis of HIV-2-infected patients receiving antiretroviral therapy (ART) is still challenging, due to the intrinsic resistance to non-nucleoside reverse transcriptase inhibitors (NNRTI) and the suboptimal response to some protease inhibitors (PI). The objective was to describe the 5-years outcomes among HIV-2 patients harboring drug-resistant viruses.

### Methods

A clinic-based cohort of HIV-2-patients experiencing virologic failure, with at least one drug resistance mutation was followed from January 2012 to August 2017 in Côte d'Ivoire. Follow-up data included death, lost to follow-up (LTFU), immuno-virological responses. The Kaplan-Meier curve was used to estimate survival rates.

### Results

A total of 31 HIV-2 patients with virologic failure and with at least one drug resistance mutation were included. Two-third of them were men, 28(90.3%) were on PI-based ART-regimen at enrolment and the median age was 50 years (IQR = 46–54). The median baseline CD4 count and viral load were 456 cells/mm$^3$ and 3.7 log$_{10}$ c/mL respectively, and the participants have been followed-up in median 57 months (IQR = 24–60). During this period, 21 (67.7%) patients switched at least one antiretroviral drug, including two (6.5%) and three (9.7%) who switched to a PI-based and an integrase inhibitor-based regimen respectively. A

**Funding:** The Main study was conducted under the IeDEA West Africa collaboration grants funded by the National Cancer Institute (NCI); Eunice Kennedy Shriver National Institute of Child Health & Human Development (NICHD); National Institute of Allergy and Infectious Diseases (NIAID); Grant number: 5U01AI069919 awarded to Pr François Dabis. The specific analyses and results reported in this publication were conducted under the EDCTP2 Training and Mobility Action awarded to Dr Boris Kevin Tchounga as Career Development Fellowship grant. Grant number: TMA 2016 CDF 1597 EDIIMark-2. The funders had no role in study design, data collection and analysis, decision to publish, or preparation of the manuscript.

**Competing interests:** The authors have no competing interest to declare.

**Abbreviations:** ART, Antiretroviral therapy; GSS, Genotypic susceptibility score; HIV-2, Human immune deficiency virus type 2; IeDEA, International epidemiologic database to evaluate AIDS; LTFU, Lost to follow up; NNRTI, Non-nucleoside reverse transcriptase inhibitors; NRTI, Nucleoside reverse transcriptase inhibitors; PI, Protease inhibitors.

total of 10(32.3%) patients died and 4(12.9%) were LTFU. The 36 and 60-months survival rates were 68.5% and 64.9%, respectively. Among the 17 patients remaining in care, six (35.3%) had an undetectable viral load (<50 c/mL) and for the 11 others, the viral load ranged from 2.8 to 5.6 $\log_{10}$ c/mL. Twelve patients were receiving lopinavir at the time of first genotype, five(42%) had a genotypic susceptibility score (GSS) $\leq$1 and 4(33%) a GSS >2.

## Conclusions

The 36-months survival rate among ART-experienced HIV-2 patients with drug-resistant viruses is below 70%,lower than in HIV-1. There is urgent need to improve access to second-line ART for patients living with HIV-2 in West Africa

## Introduction

HIV-2 is responsible for a localized AIDS epidemic that mainly affects the West African region [1–3]. The therapeutic strategy for people living with HIV-2 remain challenging, due to the intrinsic resistance of this virus to non-nucleoside reverse transcriptase inhibitors (NNRTI) and fusion inhibitors, as well as the suboptimal response to some protease inhibitors (PI) [4–6].

In absence of randomized controlled trial, there is no consensus on the therapeutic care of people living with HIV-2 [7,8]. The previous national ART guidelines of West African countries, online with WHO 2010 guidelines, recommended the initiation of a boosting lopinavir/r-based regimen as the preferred option or a three-NRTI based regimen as alternative [9]. The current British, French and USA antiretroviral (ART)-guidelines recommend initiating two NRTI associated with one boosted PI or with one integrase strand-transfer inhibitor (INSTI), and excluded the use of three NRTI as first-line regimen in patients living with HIV-2 [10–12]. More recently, the 2019 WHO guidelines recommended Dolutegravir (DTG) in combination with a nucleoside reverse-transcriptase inhibitor (NRTI) backbone as the preferred first-line regimen for people initiating ART, thus without any difference between those living with HIV-1 and HIV-2 [13]. However, the implementation of this recommendation in low- and middle-income countries, especially those with double circulation of HIV-1 and HIV-2 is ongoing slowly and DTG-based regimen were mainly prescribed to third-line patients in referral centers [14].

In case of virologic failure, HIV-2 patients were enrolled for three months in an enhanced adherence counselling program. If at the end of this period the viral load remained unsuppressed, the patient was eligible to switch treatment. In 2016, West African and European guidelines on the management of treatment failure recommended for HIV-2 patients initially receiving three NRTIs or a LP/r-based regimen as first line, to switch to darunavir (DRV) or raltegravir (RAL) / Dolutegravir (DTG) in combination with NRTI backbone [14]. With the recent recommendation of DTG as preferred first-line regardless of HIV type, the preferred second line recommended by WHO and currently endorsed by West African countries is now LP/r based regimen for HIV-2 patients [13].

Thus management of virologic failure in HIV-2 patients remain challenging despite the guidelines revision, due to the limited options (NRTI, boosted PI) available, while an increasing number of studies reported virologic failure and resistance-associated mutations to NRTIs, PIs and RAL [15–21]. These multiple resistances jeopardize the efficacy of second line HIV-2 treatment, with multidrug resistance needing boosted darunavir plus raltegravir based

regimen in resource-limited settings where HIV-2 viral load and genotypic resistance tests are neither routinely available nor affordable.

There is few data reporting experience of therapeutic care and describing long-term outcome of HIV-2-infected individuals experiencing virologic failure in resource-limited settings. Such data will be useful to orient clinicians and decision makers in the management of treatment switches in HIV-2 patients with treatment failure. This survey aimed to describe the sequence of ART regimens use, and the 5-years outcomes among HIV-2 patients harboring drug-resistant viruses in Côte d'Ivoire.

## Materials and methods

### Study design, population and settings

A clinic-based cohort study was initiated in January 2012 within the International epidemiological Database to evaluate AIDS (IeDEA) in West Africa [22]. The eligibility criteria for this cohort were assessed during a cross-sectional survey conducted to describe virologic failure and drug resistance mutations among HIV-2-infected individuals receiving ART and followed up in six HIV clinics in Abidjan, Côte d'Ivoire [15]. Based on the results of this cross-sectional survey presented elsewhere [15], adults living with HIV-2, experiencing virologic failure, and harboring at least one drug resistance mutation, were included and followed up from 2012 to 2017.

### Ethics consideration

The protocol of the IeDEA west Africa collaboration cohort was reviewed and approved in Côte d'Ivoire by the National Ethic Committee for life Science and Health (CNESVS: IORG00075). Prior to the initial enrolment in the cohort, each participant was given comprehensive information on the study protocol and procedures, and had to provide a written consent before being included.

### HIV-2 standard of care in Côte d'Ivoire

According to the national guidelines of Côte d'Ivoire at the time of study in 2012, ART was initiated in people living with HIV symptomatic stage 3–4 or asymptomatic with CD4 <350 cells/mm$^3$. In case of HIV-2 or HIV1&2, the preferred first line option was 2 NRTI plus boosted Lopinavir. Three-NRTI-based regimen was considered as alternative option (if CD4 cell counts > 200 cells/mL or Lopinavir contraindication/intolerance) [14]. In case of virologic failure, HIV-2 patients should receive the most appropriate ART regimen available, with the guidance of the national referral center for adults living with HIV (Unit of Infectious and Tropical Diseases Treichville University teaching hospital) [14].

### Study procedures

The follow-up consisted in the administration of a standardized questionnaire allowing collection of clinical (AIDS events and Non-AIDS severe morbidity), biological (CD4 count, viral load) and therapeutic (switches second line, salvage ART regimen) data, during routine follow-up visits.

### Biological procedures

A comprehensive description of the biological procedure has been published [15]. Briefly, virologic failure was defined as plasma HIV-2 RNA above 50 copies/mL using a real-time PCR assay [23]. Genotypic resistance tests (protease and reverse transcriptase sequencing) were

performed using an in-house method [15]. The interpretation was based on the HIV French resistance algorithm update of September 2017, available at http://www.hivfrenchresistance.org/index.html. A genotypic susceptibility score (GSS) was generated for each patient based on the results of genotypic analyses. In 2017, a blood sample was collected from each participant presenting at the HIV clinic, for a routine follow-up visit or returning to care after a successful tracking process. This blood samples allowed performing a viral load and an additional CD4 count.

### Outcomes and variables

The main outcomes considered were being alive, dead or LTFU. Considering LTFU like a proxy of death, a variable combining death and LTFU was defined.

Death was defined as being reported dead in the medical records of the HIV clinic or being declared dead by a close relative or a family member. A participant was considered alive if he presented for follow-up visit during the year 2017 or if he was successfully contacted during the active tracking process. The participants who did not showed up at HIV clinic for more than three months, were not known as alive, transferred out or deceased, and were not successfully tracked (phone calls and home visits when allowed in the initial consent form) were considered LTFU.

### Statistical analyses

Data analysis was conducted using STATA® version 14.0, Stata Corp, College Station, Texas USA. Kaplan Meier curve was used to estimate survival rate and Logrank test was used to compare survival between the two groups.

## Results

### Baseline characteristics

Among the 31 participants included in the study, 28 (90.3%) were receiving a PI-based regimen, 2 (6.5%) a three-NRTI-based regimen and 1 (3.2%) a raltegravir-based regimen. The median age at enrolment was 50 years (IQR = 46–54 years) and 20 patients were men (64.5%). At enrolment, the median baseline CD4 count and viral load were 456 cells/mm$^3$ (IQR = 256–751) and 3,700 c/mL (IQR = 663–7797), respectively. The initial genotypic analyses retrieved PI resistance mutations (at least one) in 26 (83.9%) participants and NRTI resistance mutations in 21 (67.7%) participants. The GSS was <2 for 14 (45.2%) patients and >2 for 10 (32.3%) others (Table 1).

### Follow up characteristics

The cumulative follow-up duration was 1327 person-months with a median duration of 52 months (IQR = 24–59). During this period, 21 (67.7%) patients switched at least one antiretroviral drug, including two (6.5%) and three (9.7%) who switched to a PI-based and an INSTI-based regimen respectively. At the censured date, 17 (55.0%) patients remained in care, while 10 (32.3%) were dead and 4 (12.9%) were LTFU. The last median CD4 count were 150 cells/mm$^3$ (IQR = 117–218) and 143 cells/mm$^3$ (IQR = 39–340) among patients dead and LTFU, respectively. Among those still in care during follow up, the 12, 36 and 60-months survival rates were 86.8%, 68.8% and 64.9% respectively (Fig 1). Neither gender (HR = 1.57, p = 0.484) nor age >50 years (HR = 0.59, p = 0.421) were associated with mortality.

Among the 17 patients remaining in care, six (35.3%) had an undetectable viral load (<50 c/mL) and for the 11 others, the median viral load was 4,334 [859–87,523] c/mL, ranging from

**Table 1. Follow-up characteristics of antiretroviral-experienced HIV-2-infected patients with identified resistance mutations from 2012 to 2017.**

| | Alive | | Death/LTFU | | Total | | p-values |
|---|---|---|---|---|---|---|---|
| | n = 17 (55%) | | n = 14 (45%) | | n = 31 | | |
| **Follow-up duration, months, Median [IQR]** | 60 [58–61] | | 19 [12–35] | | 57 [24–60] | | **0.001** |
| **Baseline CD4 count cells/mm³** | | | | | | | |
| **Median [IQR]** | 445 [266–675] | | 504 [256–855] | | 456 [256–751] | | 0.489 |
| ≤350 | 6 | (35.3) | 5 | (35.7) | 11 | (35.5) | |
| 350–500 | 4 | (23.5) | 1 | (7.2) | 5 | (16.1) | |
| ≥500 | 7 | (42.2) | 7 | (50.0) | 14 | (45.2) | |
| Missing | 0 | (0.0) | 1 | (7.1) | 1 | (3.2) | |
| **Last CD4 count cells/mm³** | | | | | | | |
| **Median [IQR]** | 281 [193–321] | | 150 [113–278] | | 230 [120–321] | | 0.202 |
| ≤350 | 13 | (76.4) | 11 | (78.7) | 24 | (77.4) | |
| 350–500 | 2 | (11.8) | 1 | (7.1) | 3 | (9.7) | |
| ≥500 | 2 | (11.8) | 1 | (7.1) | 3 | (9.7) | |
| Missing | 0 | (0.0) | 1 | (7.1) | 1 | (3.2) | |
| **Viral load at inclusion c/mL** | | | | | | | |
| **Median [IQR]** | 730 [372–4103] | | 5992 [3700–16716] | | 3700 [663–7797] | | **0.007** |
| **Viral load at closing date c/mL** | | | | | | | |
| **Median [IQR]** | 859 [0–9082] | | NA | | 859 [0–9082] | | |
| **ARV regimen at enrollment** | | | | | | | |
| PI-based | 17 | (100.0) | 11 | (78.6) | 28 | (90.3) | 0.568 |
| 3 NRTI-based | 0 | (0.0) | 2 | (14.3) | 2 | (6.5) | |
| Raltegravir-based | 0 | (0.0) | 1 | (7.1) | 1 | (3.2) | |
| **ARV drug switch (at least one)** | **12** | **(70.6)** | **9** | **(64.3)** | **21** | **(67.7)** | 0.270 |
| To darunavir* | 0 | (0.0) | 2 | (14.3) | 2 | (6.5) | |
| To raltegravir* | 2 | (11.8) | 1 | (7.1) | 3 | (9.7) | |
| **Genotypic susceptibility score** | | | | | | | 0.933 |
| <2 | 8 | (47.1) | 6 | (41.9) | 14 | (45.2) | |
| ≥2 | 5 | (29.4) | 5 | (35.7) | 10 | (32.3) | |
| Not available | 4 | (23.5) | 3 | (21.4) | 7 | (22.5) | |

IQR = interquartile range, LFTU = lost to follow-up, PI = protease inhibitor, NRTI = nucleoside reverse transcriptase inhibitor

* = non-cumulative; NA = not available; ARV = antiretroviral.

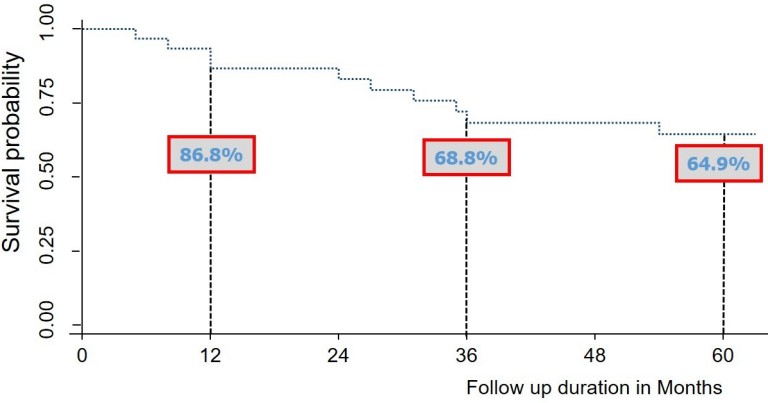

**Fig 1.**

**Table 2. Five-years ART-response among HIV-2 infected patients surviving after experiencing treatment failure with identified drug resistance mutations in Côte d'Ivoire, West Africa.**

| Patients demographics | | | ART Response in 2012 | | | Genotypic analyses in 2012 | | | ART Response in 2017 | | |
|---|---|---|---|---|---|---|---|---|---|---|---|
| ID | Age (years) | Sex | ART regimen at treatment failure | CD4 (/mm³) | VL (c/mL) | NRTI Mutations | PI Mutations | GSS | Last ART regimen | CD4 (/mm³) | VL (c/mL) |
| PAT-S01 | 51 | M | AZT/3TC/LPV/r | 106 | 340 | M184V | V47A, V62A | 1 | ABC/3TC/LPV/r | 281 | 859 |
| PAT-S02 | 53 | M | ABC/3TC/LPV/r | 919 | 713 | M184V | I54M | 2 | TDF/3TC/LPV//r | 209 | 4,334 |
| PAT-S03 | 56 | M | AZT/3TC/LPV/r | 675 | 3085 | M184V, S215Y | V47A, I82M, L90M | 0 | AZT/3TC/LPV/r | 351 | 372,346 |
| PAT-S04 | 30 | F | AZT/3TC/LPV/r | 119 | 663 | NA | V47A, | NA | AZT/3TC/LPV/r | 291 | 87,523 |
| PAT-S05 | 58 | M | AZT/3TC/LPV/r | 545 | 149 | NA | Wild-Type | NA | TDF/3TC/LPV/r | 362 | - |
| PAT-S06 | 60 | F | ABC/ddI/LPV/r | 791 | 4103 | M184V | NA | NA | TDF/3TC/LPV/r | 79 | 13,457 |
| PAT-S07 | 33 | M | SQV/LPV/r | 141 | 33287 | Q151M | I50V, I54M, L90M | 0 | TDF/3TC/DRV/r/RAL | 214 | - |
| PAT-S08 | 50 | M | ABC/3TC/SQV/r | 425 | 8790 | Q151M, K65R, K70R, M184V, K223R | V47A, I50V, I84V, L90M, L99F | 0 | TDF/3TC/DRV/r/RAL | 69 | - |
| PAT-S09 | 49 | M | ddI/ABC/LPV/r | 751 | 1471 | M184V | V47A, L99F | 2 | TDF/3TC/DRV/r | 256 | 1,535 |
| PAT-S10 | 54 | M | AZT/3TC/LPV/r | 357 | 5156 | Wild-Type | Wild-Type | 3 | AZT/3TC/LPV/r | 103 | 2,337 |
| PAT-S11 | 60 | M | ABC/ddI/LPV/r | 445 | 3016 | Q151M, M184V, S215F | I54M, I84V | 0 | ABC/ 3TC/ TDF/ DRV/r | 193 | 9,082 |
| PAT-S12 | 50 | M | TDF/FTC/LPV/r | 606 | 193 | M184I | Wild-Type | 2 | ABC/3TC/LPV/r | 120 | - |
| PAT-S13 | 53 | M | AZT/3TC/LPV/r | 157 | 395 | M184V | V47A | 1 | AZT/3TC/LPV/r | 297 | 223,827 |
| PAT-S14 | 43 | F | TDF/FTC/LPV/r | 336 | 91 | M184V | Wild-Type | 2 | TDF/3TC/LPV/r | 774 | - |
| PAT-S15 | 55 | M | AZT/3TC/LPV/r | 712 | 372 | NA | V47A | NA | AZT/3TC/DRV/r | 855 | - |
| PAT-S16 | 53 | M | AZT/3TC/LPV/r | 266 | 730 | M184V, S215Y | V47A, V62AI | 0 | AZT/3TC/LPV/r | 321 | 584 |
| PAT-S17 | 49 | F | TDF/FTC/LPV/r | 466 | 5146 | K65R, N69S, M184V | V47A | 0 | TDF/3TC/LPV/r | 292 | 685 |

NA: Not amplified; ID = participant identifier; ART = antiretroviral therapy; VL = viral load; NRTI = Nucleoside reverse transcriptase inhibitors; PI = protease inhibitors; GSS = genotypic susceptibility score

584 to 372,346 c/mL (Table 2). Twelve of these patients were receiving lopinavir at time of first genotyping analysis, eight (47.1%) had a GSS <2 and five (29.4%) a GSS >2. Their last median CD4 count was 281 cells/mm³ (IQR = 209–351), not significantly different to those who died or were LTFU (p = 0.20). none of participants CD4 increased after switch or after the first virologic failure.

Among the 10 patients declared dead, 6 (60.0%) had CD4 count <200 cells/mm³, 7 (70.0%) had a boosted PI in their last known ART regimen, five (50.0%) had a GSS <2 and 7 (70.0%) changed at least one drug in their ARV regimen at least once after the diagnosis of drug resistance mutation. Regarding treatment, 5 of them switched to an unappropriated and non-effective ART regimen according to the genotypic resistance test. three of them were maintained on boosted-lopinavir as a compassionate treatment until they died. Two of them received a

non-recommended regimen for HIV-2 based on Atazanavir although LPV/r-based regimen was effective.

## Discussion

In this observational cohort study conducted in Côte d'Ivoire, we reported after 5 years of diagnosis of at least one drug resistance mutation to first-line ART regimen among HIV-2 patients, high mortality and lost to follow-up. Half of HIV-2 patients were not retained in care 5 years after the diagnosis of drug resistance mutation.

This high rate of mortality could be explained by the lack of access to second-line therapy for these patients in resource-limited settings. According to the US and French ART guidelines, only integrase inhibitors such as raltegravir, elvitegravir or dolutegravir or the CCR5 antagonist maraviroc (MVC) could be used as a second-line therapy among HIV-2 patients [24–28]. However, in most west African countries, these antiretroviral molecules are still scarcely available. In our study only three (9.3%) patients diagnosed with drug resistant viruses switched to the INSTI raltegravir, none of them received dolutegravir. Among the patients dead or LTFU, only one (7.1%) and two (14.2%) switched to an INSTI and to the PI darunavir, respectively. For these latter patients, the median last known CD4 count was 150 cells/mm$^3$, indicating that more than half of them were in advanced HIV disease according to the immunologic definition [29]. Increase the availability of more efficient antiretroviral drugs like integrase inhibitors is critical for patients living with HIV-2 for whom the therapeutic arsenal is limited specifically for patients harboring viruses with drug resistance mutations [15].

In resource-limited settings, there was no clear sequence of ART regimen use for the treatment of HIV-2-infected individual experiencing virologic failure [14]. This lack of clear guidelines may be the consequence of limited data on the switch of treatment among HIV-2-infected patients, the lack of routine implementation of validated tool for viral load monitoring, the absence of definition of immunological failure and the paucity of data regarding drug resistance mutations [7,8,23,30–32]. Since the mortality remains high among HIV-2 patients receiving ART [33,34], it is critical to address all the gaps and need in terms of data in order to propose clear guidelines for the treatment of HIV-2 patients experiencing virologic failure.

In the present study, six of the 17 patients remaining in care had no active antiretroviral drug in their regimen. In fact, salvage regimen for those patients with a good GSS should use boosted-darunavir and DTG which remains active in some cases of resistance to the first generation of INSTI [21,35]. Thus, in this context, genotypic resistance tests are needed in order to prevent DTG from being functional monotherapy which will result to the selection of DTG resistances. Unfortunately, according to the GSS of the patients of the present study it seems that DTG will not be sufficient and additional new drugs are needed, such as maraviroc or broad-spectrum neutralizing antibodies (Ibalizumab) [36–38]. Unfortunately, in our study population, among the 10 patients with no active ARV drug (GSS = 0), four died, arguing for the need of new therapeutic options for HIV-2 infection.

Since 2018, WHO guidelines recommend tenofovir plus lamivudine plus dolutegravir based regimen as the preferred first-line option according to expert opinion, pilot studies and in vitro data [39]. This will change the management of HIV-2-infected patients with the use of boosted-PI such as lopinavir or darunavir with an optimized NRTI-backbone in second-line [21,27,39–41]

### Study limitations and strengths

Although the study was conducted in the HIV clinics with a proper documentation of patient's follow-up, the main limitation is the lack of documentation of the cause of death. For all the

patients who were not reported dead at the HIV clinic, it was not possible to identify the clinical cause of death, as the only information available in the official death certificate was "disease". In addition, viral load measurement was not routinely performed and only two measures (2012 and 2017) were available for the analysis, making it impossible to describe the evolution according to the switches of antiretroviral drugs during the five years of follow-up. Furthermore, this study presents data of a small population and the estimates may suffer from lack of statistic power. However, to our knowledge, this is one of the first report on treatment outcomes among HIV-2 patients who experienced virologic failure with at least one drug resistance mutation in West Africa. Data on long-term follow-up among HIV-2 patients are also limited and this study highlights the challenge to determine the sequence of ART use in this population.

## Conclusions

Our data call for the urgent access to second-line and third-line therapy among HIV-2 patients. Clinical trials for HIV-2-infected patients harboring multi-drug resistant viruses should be conducted in both resource-limited settings and western countries. Results from the first randomized controlled trial on HIV-2 (FIT-2) Expected in 2020, will be helpful to define a sequence of ART initiation among HIV-2 patients.

## Acknowledgments

We would like to warmly thank Drs Denise Dekpanou; Albert Minga; Eugène messou; Zelika Diallo and Serge Kooley for their contribution during the implementation of the study, as well as Mr Azany Jean Claude for his strong support in data management and study monitoring and Dr Boni Simon for his support to the clinical monitoring and critical review of the manuscript.

## Author Contributions

**Conceptualization:** Boris K. Tchounga, Didier K. Ekouevi.

**Data curation:** Boris K. Tchounga, Charlotte Charpentier, Serge P. Eholie, Didier K. Ekouevi.

**Formal analysis:** Boris K. Tchounga, Charlotte Charpentier, Didier K. Ekouevi.

**Funding acquisition:** Boris K. Tchounga, François Dabis, Didier K. Ekouevi.

**Investigation:** Charlotte Charpentier, François Dabis, Serge P. Eholie, Didier K. Ekouevi.

**Methodology:** Boris K. Tchounga, Patrick A. Coffie, François Dabis, Diane Descamps, Serge P. Eholie, Didier K. Ekouevi.

**Project administration:** Boris K. Tchounga, Didier K. Ekouevi.

**Supervision:** Boris K. Tchounga, Diane Descamps, Serge P. Eholie, Didier K. Ekouevi.

**Validation:** Boris K. Tchounga, Charlotte Charpentier, Patrick A. Coffie, Diane Descamps, Serge P. Eholie, Didier K. Ekouevi.

**Visualization:** Patrick A. Coffie, François Dabis, Diane Descamps, Didier K. Ekouevi.

**Writing – original draft:** Boris K. Tchounga, Didier K. Ekouevi.

**Writing – review & editing:** Boris K. Tchounga, Charlotte Charpentier, Patrick A. Coffie, François Dabis, Diane Descamps, Serge P. Eholie.

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
