## [Decision Letter · Decision Letter 0]

22 Jan 2020

PONE-D-19-33514

Survival among antiretroviral-experienced HIV-2 patients experiencing virologic failure with drug resistance mutations in Cote d’Ivoire West Africa

PLOS ONE

Dear Dr Tchounga,

Thank you for submitting your manuscript to PLOS ONE. After careful consideration, we feel that it has merit but does not fully meet PLOS ONE’s publication criteria as it currently stands. Therefore, we invite you to submit a revised version of the manuscript that addresses the points raised during the review process.

We would appreciate receiving your revised manuscript by Mar 07 2020 11:59PM. To enhance the reproducibility of your results, we recommend that if applicable you deposit your laboratory protocols in protocols.io, where a protocol can be assigned its own identifier (DOI) such that it can be cited independently in the future. For instructions see: http://journals.plos.org/plosone/s/submission-guidelines#loc-laboratory-protocols

We look forward to receiving your revised manuscript.

Kind regards,

Cristian Apetrei, MD, PhD

Academic Editor

PLOS ONE

Journal Requirements:

4.One of the noted authors is a group or consortium IeDEA West Africa. In addition to naming the author group, please list the individual authors and affiliations within this group in the acknowledgments section of your manuscript. Please also indicate clearly a lead author for this group along with a contact email address.

Reviewers' comments:

Reviewer's Responses to Questions

**Comments to the Author**

1. Is the manuscript technically sound, and do the data support the conclusions?

Reviewer #1: Yes

Reviewer #2: Yes

2. Has the statistical analysis been performed appropriately and rigorously? 

Reviewer #1: No

Reviewer #2: Yes

3. Have the authors made all data underlying the findings in their manuscript fully available?

Reviewer #1: Yes

Reviewer #2: Yes

4. Is the manuscript presented in an intelligible fashion and written in standard English?

Reviewer #1: Yes

Reviewer #2: Yes

5. Review Comments to the Author

Reviewer #1: The analysis is based on 5–year outcomes of HIV-2 infected patients harboring drug resistance mutations in Cote d'Ivoire. This study included a small number of patients, compared with similar studies recently published. Many publications debated on this subject, most of them focusing on HIV-1 patients, but only a few articles discussed HIV-2 patients and the detection of drug resistance mutations.

A major concern is the lack of statistical comparison in table 1 between the two categories of patients (alive and death) (ex. chi-squared test). Thus, the study has a practical utility, as it fills in information gaps concerning disease evolution under ARV therapy in this category of patients in resource-limited settings but it needs some revisions.

Reviewer #2: The study titled “Survival among antiretroviral-experienced HIV-2 patients experiencing virologic failure with drug resistance mutations in Cote d’lvoire West Africa” describes the various treatment modalities received by patients infected with HIV-2 experiencing virologic failure in six HIV clinics in Cote d’lvoire and the 5-year outcome of these HIV-2 patients harboring drug-resistant viruses. This study stresses on the need for clear treatment guidelines for virologic failure in HIV-2 infection and calls for an increased access to second-line and third-line drugs for HIV-2 patients in resource-limited settings.

Comments

1. In the abstract, lines 61 and 62, the authors have mentioned that “The survival rate of HIV-2 ART-experienced patients with drug resistant viruses is somewhat low.” Please accurately describe “somewhat low” based on the study’s results. Also, please mention what this is in comparison to.

2. The introduction overall is very short and not detailed. The following comments pertain to the introduction:

a. In the second paragraph, the authors talk about the current British, French and USA ART guidelines but for comparison with West African and WHO guidelines, the authors have utilized previous West African and 2010 WHO guidelines. Please include the current WHO and West African ART guidelines here.

b. In the next paragraph, the authors go on to conclude that “ARV optimal combination in case of virologic failure is not clear.” Please elaborate on what the current guidelines are regarding virologic failure are before moving on to discuss the lack of clarity of the guidelines.

c. In line 92, it is unclear what the authors mean by “sequencing of ART regimens”. Please clarify the same in all the subsequent places where this is mentioned.

d. The authors may consider describing further the importance of this study and how this study may impact the field.

3. In the results section, in lines 179 and 180, the authors observed that among the 17 patients, “6 had undetectable viral loads (<50 c/mL)”. However, the authors go on to note that the viral loads “for the 11 others” (implying that these 11 patients had detectable viral loads) range from “0 to 372,346 c/mL”. This is misleading as undetectable viral loads are described as <50 c/mL but the viral loads of the 11 patients that were detected ranged starting from 0 c/mL. Please clarify.

4. In lines 225-228, the authors declare certain factors to be the cause for the lack of clear guidelines for the treatment of HIV-2 infected individuals experiencing virologic failure. Since these factors have not been evaluated in this study, please cite a reference or indicate how the authors reached to this conclusion. If this is a speculation and the authors are calling for further studies to evaluate these factors, please indicate so appropriately.

5. In line 233, the authors have stated that 5 patients had no active antiretroviral drug in their regimen. However, in line 241, the authors have stated “4 out of 6 patients with no active ARV drug have died.” Please clarify whether it is 5 or 6 patients who had no active antiretroviral drugs in their regimen.

6. The authors may consider citing outcome in HIV-2 infected patients who do not harbor drug resistant viruses, for comparison, in order to make their results more meaningful.

7. There are numerous minor grammatical errors throughout the paper. Some sentences do not convey their intended meaning. Please have these edited.

6. PLOS authors have the option to publish the peer review history of their article (what does this mean?). If published, this will include your full peer review and any attached files.

Reviewer #1: No

Reviewer #2: No

---

## [Author Response · Author response to Decision Letter 0]

6 Jul 2020

Reviewer #1: 

The analysis is based on 5–years outcomes of HIV-2 infected patients harboring drug resistance mutations in Cote d'Ivoire. This study included a small number of patients, compared with similar studies recently published. Many publications debated on this subject, most of them focusing on HIV-1 patients, but only a few articles discussed HIV-2 patients and the detection of drug resistance mutations.

A major concern is the lack of statistical comparison in table 1 between the two categories of patients (alive and death) (ex. chi-squared test). Thus, the study has a practical utility, as it fills in information gaps concerning disease evolution under ARV therapy in this category of patients in resource-limited settings but it needs some revisions.

Author’s response: We thank the reviewer for this encouraging comment regarding the practical utility of the study. Despite the low population size, we perform krouska wallis test to compare medians and Fischer Exact test to compare means and provide the p-values in table 1 in the revised manuscript. 

Reviewer #2: 

The study titled “Survival among antiretroviral-experienced HIV-2 patients experiencing virologic failure with drug resistance mutations in Cote d’lvoire West Africa” describes the various treatment modalities received by patients

infected with HIV-2 experiencing virologic failure in six HIV clinics in Côte d’lvoire and the 5-years outcome of these HIV-2 patients harboring drug-resistant viruses. This study stresses on the need for clear treatment guidelines for virologic failure in HIV-2 infection and calls for an increased access to second-line and third-line drugs for HIV-2 patients in resource-limited 

Comments

1. In the abstract, lines 61 and 62, the authors have mentioned that “The survival rate of HIV-2 ART-experienced patients with drug resistant viruses is somewhat low.” Please accurately describe “somewhat low” based on the study’s results. Also, please mention what this is in comparison to.

Author’s response: We thank the reviewer for this comment, we reviewed the sentence and specified the comparison factor as follows: 

The 36-months survival rate among ART-experienced HIV-2 patients with drug-resistant viruses is below 70%, lower than in HIV-1.

2. The introduction overall is very short and not detailed. The following comments pertain to the introduction:

a. In the second paragraph, the authors talk about the current British, French and USA ART guidelines but for comparison with West African and WHO guidelines, the authors have utilized previous West African and 2010 WHO guidelines. Please include the current WHO and West African ART guidelines here.

Author’s response: We thank the reviewer for this comment; we reviewed the section to include the current WHO and West African ART guidelines here as follows: 

More recently, the 2019 WHO guidelines recommended Dolutegravir (DTG) in combination with a nucleoside reverse-transcriptase inhibitor (NRTI) backbone as the preferred first-line regimen for people initiating ART, thus without any difference between those living with HIV-1 and HIV-2 (13).. However, the implementation of this recommendation in low- and middle-income countries, especially those with double circulation of HIV-1 and HIV-2 is ongoing slowly and DTG-based regimen were mainly prescribed to third-line patients in referral centers (14). 

b. In the next paragraph, the authors go on to conclude that “ARV optimal combination in case of virologic failure is not clear.” Please elaborate on what the current guidelines are regarding virologic failure are before moving on to discuss the lack of clarity of the guidelines.

Author’s response: We thank the reviewer for this comment. We reviewed the section to elaborate on what the current guidelines state regarding virologic failure in the study setting and we clarified the sentence regarding clarity of guidelines. 

In case of virologic failure, HIV-2 patients were enrolled for three months in an enhanced adherence counselling program. If at the end of this period the viral load remained unsuppressed, the patient was eligible to switch treatment. In 2016, West African and European guidelines on the management of treatment failure recommended for HIV-2 patients initially receiving three NRTIs or a LP/r-based regimen as first line, to switch to darunavir (DRV) or raltegravir (RAL) / Dolutegravir (DTG) in combination with NRTI backbone (14). With the recent recommendation of DTG as preferred first-line regardless of HIV type, the preferred second line recommended by WHO and currently endorsed by West African countries is now a LP/r based regimen for HIV-2 patients (13). 

c. In line 92, it is unclear what the authors mean by “sequencing of ART regimens”. Please clarify the same in all the subsequent places where this is mentioned.

Author’s response: We thank the reviewer for this comment, highlighting this typo. We wanted to express the idea that the sequence of use of available ART regimens for HIV-2 patients was not clearly defined and thus admitted by the clinicians and researchers. The typo introduces a confusion between sequencing that applies for the biological genotyping procedure and the sequences meaning the treatment steps during the process of ART management. We reviewed the updated version of the manuscript to use sequencing and sequence when appropriated. 

d. The authors may consider describing further the importance of this study and how this study may impact the field.

Author’s response: We thank the reviewer for this comment. We have added in the introduction a sentence describing how this study may impact the field, especially the clinical practice in African countries where HIV-2 circulates.

3. In the results section, in lines 179 and 180, the authors observed that among the 17 patients, “6 had undetectable viral loads (<50 c/mL)”. However, the authors go on to note that the viral loads “for the 11 others” (implying that these 11 patients had detectable viral loads) range from “0 to 372,346 c/mL”. This is misleading as undetectable viral loads are described as <50 c/mL but the viral loads of the 11 patients that were detected ranged starting from 0 c/mL. Please clarify.

Author’s response: We thank the reviewer for this comment. We reviewed the numerical figures and corrected the mistake. This was corrected in the updated version of the manuscript as follows: 

Among the 17 patients remaining in care, six (35.3%) had an undetectable viral load (<50 c/mL) and for the 11 others, the median viral load was 4,334 [859 - 87,523] c/mL, ranging from 584 to 372,346 c/mL (Table 2).

4. In lines 225-228, the authors declare certain factors to be the cause for the lack of clear guidelines for the treatment of HIV-2 infected individuals experiencing virologic failure. Since these factors have not been evaluated in this study, please cite a reference or indicate how the authors reached to this conclusion. If this is a speculation and the authors are calling for further studies to evaluate these factors, please indicate so appropriately.

Author’s response: We thank the reviewer for this comment. The factors cited were illustrated with appropriate references inserted in the manuscript. We also reviewed the sentence to modulate the role of these factors as we did not evaluate them ourselves. 

5. In line 233, the authors have stated that 5 patients had no active antiretroviral drug in their regimen. However, in line 241, the authors have stated “4 out of 6 patients with no active ARV drug have died.” Please clarify whether it is 5 or 6 patients who had no active antiretroviral drugs in their regimen.

Author’s response: We acknowledge that the two sentences were a little bit confusing. Among the 17 patients still alive by the time of this evaluation, six had a GSS =0, meaning that they had no active drug in their regimen. In the second sentence we were highlighting the fact that in our study population, 10 out of the 31 patients with virologic failure had no effective drug in their regimen (GSS=0), and among these 10 patients, four died during the 5 years of follow up. The paragraph was updated to clarify this in the revised version of the manuscript as follows: 

Unfortunately, in our study population, among the 10 patients with no active ARV drug (GSS=0), four died, arguing for the need of new therapeutic options for HIV-2 infection.

6. The authors may consider citing outcome in HIV-2 infected patients who do not harbor drug resistant viruses, for comparison, in order to make their results more meaningful.

Author’s response: We acknowledge the importance of a comparison with HIV-2 clients who do not harbor drug resistant viruses. We were unable to find comparable studies presenting survival or mortality data in HIV-2 patients according to the existence or the absence of viral resistance mutation after a long period of follow up. Most studies did not perform systematic viral load and resistance mutation testing for all the patients enrolled. We have taken good note of this comment and will consider the possibility of an ancillary study to document the outcome of the patients who did not harbor drug resistant viruses. We did not amend the revised version of the manuscript according to this comment as this sub--study is not planned yet. 

7. There are numerous minor grammatical errors throughout the paper. Some sentences do not convey their intended meaning. Please have these edited.

Author’s response: We tried our best to improve the English and the style of the manuscript with the assistance of a native English speaker.

---

## [Editor Report · Decision Letter 1]

13 Jul 2020

Survival among antiretroviral-experienced HIV-2 patients experiencing virologic failure with drug resistance mutations in Cote d’Ivoire West Africa

PONE-D-19-33514R1

Dear Dr. Tchounga,

We’re pleased to inform you that your manuscript has been judged scientifically suitable for publication and will be formally accepted for publication once it meets all outstanding technical requirements.

Kind regards,

Cristian Apetrei, MD, PhD

Academic Editor

PLOS ONE
---

## [Editor Report · Acceptance letter]

24 Jul 2020

PONE-D-19-33514R1 

Survival among antiretroviral-experienced HIV-2 patients experiencing virologic failure with drug resistance mutations in Cote d’Ivoire West Africa 

Dear Dr. Tchounga:

I'm pleased to inform you that your manuscript has been deemed suitable for publication in PLOS ONE. Congratulations! Your manuscript is now with our production department. 

Kind regards, 

on behalf of

Dr. Cristian Apetrei 

Academic Editor

PLOS ONE